# When Does Causal Regularization Help?
# A Systematic Study of Boundary Conditions in Spurious Correlation Learning

## Abstract

We challenge the conventional wisdom that explicit causal regularization is universally necessary for out-of-distribution generalization. Through systematic investigation on ColoredMNIST, we discover that reconstructive architectures like autoencoders provide a powerful **implicit causal bias** that can reduce the need for explicit methods like IRM or HSIC. Autoencoder baselines achieve 82-86% accuracy with 99% spurious correlation, with explicit causal losses adding only marginal (0-4pp) gains.

However, using the Atlasing Pattern Space (APS) framework—a modular toolkit combining topology preservation (T), causal invariance (C), and energy shaping (E)—we establish clear **boundary conditions** for when explicit regularization is essential. Our experiments across multiple domains reveal that: (1) explicit causal methods become critical when architectural bias is absent or spurious correlations are pathologically strong (approaching 100%); (2) topology preservation improves kNN fidelity in high-dimensional vision tasks but fails in low-dimensional synthetic settings; and (3) energy-based regularization effectively prevents overfitting.

Through controlled experiments including a systematic study of component domain-specificity, we demonstrate that regularization components are not universally beneficial but rather require careful domain-specific validation. Our results reframe causal learning as a hierarchical process: architectural choice is primary, with explicit regularizers serving as targeted, domain-specific corrections when architectural bias proves insufficient. All main benchmark results are mean $\pm$ std over $N$ seeds (see Appendix); parameter sweeps (Table 4) use a single seed. We evaluate robustness primarily via Worst-Group Accuracy on group-labeled benchmarks.

## 1 Introduction

The pursuit of models that generalize out-of-distribution (OOD) has centered on developing explicit causal regularization techniques like Invariant Risk Minimization (IRM) Arjovsky et al. (2019) and Hilbert-Schmidt Independence Criterion (HSIC) losses Gretton et al. (2005). The underlying assumption is that standard models naively exploit spurious correlations, and that these explicit regularizers are necessary to force models to learn invariant, causal features.

This paper challenges that assumption. We demonstrate that for a broad class of models, **architectural choice can be a more powerful driver of causal learning than explicit regularization**. Specifically, we find that reconstructive models like autoencoders possess a strong **implicit causal bias**. By being forced to reconstruct the input, these models naturally learn to prioritize structural (causal) features over superficial (spurious) ones, even when the spurious features are overwhelmingly correlated with the label.

Our central finding, derived from a systematic study on ColoredMNIST, is that a standard autoencoder baseline achieves 82-86% accuracy with 99% spurious correlation. Explicit causal regularizers, when added, provide only marginal gains (0-4pp), suggesting they are largely redundant when a strong architectural bias

is already present. This discovery reframes the central question from *'How do we add causal constraints?'* to *'When and why are they actually needed?'*

To dissect this interplay between implicit and explicit bias, we employ the **Atlasing Pattern Space (APS)** framework as a modular diagnostic toolkit. APS combines three regularizers:

- **Topology (T)**: Preserves the manifold structure of the data.

- **Causality (C)**: Enforces invariance to nuisance factors (e.g., via HSIC).

- **Energy (E)**: Shapes the latent space using a data-driven energy function.

Using APS, we establish clear **boundary conditions** for causal learning. We show that explicit methods (like the C component) become critical only when the implicit architectural bias is absent (e.g., in simple feed-forward classifiers) or when the data presents pathological spurious correlations (approaching 100%).

Beyond the architectural bias finding, our systematic experiments across vision and NLP domains reveal domain-specific boundary conditions for each APS component. We find that topology preservation improves kNN fidelity in high-dimensional vision tasks (MNIST) but fails to provide measurable benefits in low-dimensional synthetic settings, demonstrating that geometric regularization is not universally applicable. Energy-based regularization consistently prevents overfitting but provides only marginal OOD accuracy improvements, suggesting its role is primarily as a capacity control mechanism rather than a causal learning tool.

Our contributions are fourfold:

- **Implicit Causal Bias**: We identify the intrinsic causal bias in autoencoders that provides robustness to spurious correlations.

- **APS Framework**: We unify topology, causality, and energy into a modular diagnostic framework.

- **TopologyEnergy**: We propose a novel energy mechanism that aligns with topological constraints, resolving failures of memory-based approaches.

- **Boundary Conditions**: We establish a systematic decision framework for when explicit regularization is actually necessary.

Ultimately, this work advocates for a more nuanced, hierarchical approach to causal learning: begin with the right architecture, and only then apply explicit regularization as a targeted, second-order correction.

**Motivation:** The name *"Atlasing"* evokes the creation of a map or atlas of all patterns (e.g. linguistic or visual patterns) such that distance and neighborhoods on the map reflect true semantic or functional similarity. Unlike standard embedding methods which largely treat latent dimensions as unstructured, APS treats representation learning as a **manifold learning problem** with additional causal and energy-based regularization. By doing so, APS aims to produce latent "charts" that are easier to interpret and navigate – much like an atlas that faithfully represents the terrain: - In NLP, an APS-learned embedding might place synonyms or contextually similar phrases in adjacent regions (topology), align dimensions with abstract concepts (causality), and form energy basins for distinct topics or themes (energy). - In computer vision, APS could map images such that images with similar content or style cluster together (topology), latent variables isolate factors like lighting or viewpoint (causality), and each object category corresponds to an energy basin that stores its prototypical patterns. - In recommendation systems, user/item embeddings could be structured so that similar users/items lie in contiguous latent neighborhoods, confounding factors (e.g. popularity) are factored out, and communities or genres appear as attraction basins.

By integrating these properties, APS promises representations that support **better generalization** (through invariant features), **robustness to spurious correlations** (through causal structure), and **enhanced interpretability** (through topologically and energetically organized latent maps). In the following sections, we formalize the APS framework and discuss related work that inspires each component (Topology, Causality, Energy). We then outline the methodology for implementing APS and propose experiments to evaluate its benefits.

## 2 Related Work

### 2.1 Topology-Preserving Embeddings

Our emphasis on latent **topology preservation** builds on a rich history of manifold learning and neighbor-preserving embeddings. Classical techniques like **t-SNE** (Van der Maaten & Hinton, 2008) and **UMAP** (McInnes et al., 2018) aim to embed high-dimensional data into low dimensions (e.g. 2D) for visualization, such that similar points stay close and multi-scale structure is maintained. In particular, UMAP uses a framework from algebraic topology to learn a low-dimensional mapping that preserves both local and some global structure of the data manifold (McInnes et al., 2018), while t-SNE focuses on retaining local neighbor affinities and revealing cluster structure at multiple scales (Van der Maaten & Hinton, 2008). These methods underscore the value of respecting the intrinsic topology of data, although they are typically used as post-hoc visualizers rather than as trainable model components.

In neural network research, recent work has explicitly added topological or geometric constraints to latent spaces. **Topological Autoencoders** (Moor et al., 2020) introduced a differentiable loss based on persistent homology to ensure that the topology (e.g. connectivity, loops) of the latent space matches that of the input space. By penalizing differences in Betti numbers and other topological features between input and latent distributions, they preserved multi-scale connectivity and improved interpretability of latent dimensions. Other approaches enforce local geometric fidelity: for example, **Local Distance Preserving Autoencoders** (Chen et al., 2022) add a loss that keeps the distances between each point and its $k$-nearest neighbors in data space similar in latent space. This is achieved via a continuous $k$-NN graph that captures topological features at all scales, used as a constraint during training. Such methods align with earlier ideas like **Laplacian eigenmaps** and **locally linear embedding (LLE)**, which also preserve neighbor relations in a lower-dimensional embedding of the data manifold. We will revisit this connection in Section 3.1, where we define our proposed TopologyEnergy as a differentiable relaxation based on continuous k-NN similarity.

Graph-based regularization of latent geometry has shown promise in autoencoders. For instance, **Neighborhood Reconstructing Autoencoders (NRAE)** (Lee et al., 2021) incorporate a term ensuring that each data point's local neighborhood (from a precomputed graph) is reconstructed by the decoder, thus correcting "wrong local connectivity and geometry" often observed in vanilla AEs. Similarly, the **Witness Autoencoder (W-AE)** (Schönenberger et al., 2020) and **Geometry-Regularized Autoencoder (GRAE)** (Duque et al., 2020) introduced topological and geometric regularizers (e.g. using witness complexes or manifold charts) to shape the latent space. These works demonstrate that **imposing topology-awareness during representation learning leads to latent spaces that better reflect the true structure of data**, which can improve downstream tasks and the realism of interpolations. APS adopts this principle: our **Topology (T)** component will preserve neighborhood relationships (e.g. via a $k$-NN graph or topological loss) so that the learned atlas maintains the continuity and connectivity of the original pattern space.

### 2.2 Causal and Invariant Representation Learning

The **Causality (C)** component of APS seeks to make latent features invariant to nuisance factors and aligned with stable, meaningful properties. This idea is inspired by research in **causal representation learning** and **domain generalization**. A key insight from causality is that models should capture the *invariant mechanisms* underlying data rather than spurious correlations. **Invariant Risk Minimization (IRM)** (Arjovsky et al., 2019) formalized this by learning a data representation such that *the optimal classifier on that representation is the same across multiple environments*. By leveraging data from different environments (or domains), IRM encourages the encoder to discard features that are inconsistent (spurious) and keep those that have a stable relationship with the target, thereby improving out-of-distribution (OOD) generalization. APS can incorporate this principle by using multiple data contexts or augmentations and adding a penalty if a classifier's predictions differ between contexts when using the APS embedding.

Another line of work uses **independence criteria** to enforce invariances. The *Hilbert-Schmidt Independence Criterion* (HSIC) is a kernel-based measure of statistical independence. It has been used as a loss to encourage representations $Z$ to be independent of certain variables $V$ (for example, sensitive attributes or domain labels). Greenfeld & Shalit (2020) applied HSIC as a regularizer to achieve robust models under

covariate shift. By penalizing any dependence between the model's residuals and the input distribution, their HSIC-based loss yielded predictors where $Y - \{\hat{f}\}(X)$ is nearly independent of $X$, corresponding to a scenario where only the causal relation (and independent noise) remains. In APS, we can use HSIC-based penalties to encourage that the learned latent $Z$ is independent of nuisance factors (e.g. style, noise, context that we want to factor out). Similarly, other works like *Domain-Adversarial Training* and *Maximum Mean Discrepancy (MMD)* have sought to remove domain-specific information from embeddings, but HSIC offers a direct, differentiable independence measure.

There is also overlap between invariant representation learning and **disentangled representation learning**. Methods such as $\beta$**-VAE** (Higgins et al., 2017) aim to learn latent factors that correspond to independent generative factors of variation (Higgins et al., 2017). By constraining the VAE's latent channel capacity (via a higher $\beta$ weight on the KL-divergence term), $\beta$-VAE encourages the latent dimensions to capture distinct aspects of the data (for example, in an image dataset, one dimension may capture "rotation" while another captures "scale") (Higgins et al., 2017). The result is an interpretable factorized representation that is aligned with *causal factors* in the data generation process, achieved without supervision. APS's causality module shares this goal of **isolating meaningful factors**: through losses like IRM or HSIC (and potentially by borrowing ideas from $\beta$-VAE to enforce factorization), APS encourages each latent dimension or subspace to correspond to a stable property of the input, invariant to minor changes or context. Indeed, the broader vision of **causal representation learning** is to uncover latent features that correspond to real-world causal variables, a direction articulated in surveys like Schölkopf et al. (2021). APS contributes to this direction by integrating causal invariance constraints directly into the representation learning objective.

## 2.3 Energy-Based Models and Attractor Networks

The **Energy (E)** component of APS introduces an **energy-based perspective** to the latent space. Energy-Based Models (EBMs) assign an unnormalized "energy" score to configurations (in our case, latent vectors), such that low-energy regions correspond to probable or familiar patterns. By shaping the latent space's energy landscape into **basins of attraction**, APS aims to create distinct wells (valleys) that capture clusters or prototypes of patterns. This idea is reminiscent of **Hopfield networks** and other attractor models. A classical Hopfield network (Hopfield, 1982) stores patterns as stable fixed points of a dynamical system; when the network state is perturbed to a new input, it iteratively updates and converges to the nearest stored pattern (an attractor). Recent work has modernized this concept: *"Hopfield Networks is All You Need"* (Ramsauer et al., 2021) showed that a continuous-state Hopfield layer can store exponentially many patterns and that its update rule is equivalent to the Transformer's attention mechanism (Ramsauer et al., 2021). Importantly, they identified different types of energy minima in such networks: global minima that average over all patterns, metastable states averaging subsets of patterns, and fixed-point attractors corresponding to individual stored patterns (Ramsauer et al., 2021). This suggests that deep networks can incorporate Hopfield-like memory to perform pooling, association, and rapid content-based retrieval (Ramsauer et al., 2021). APS leverages this concept by aiming for a latent space where each significant pattern or concept acts as an **attractor**. For example, in an NLP context, an abstract concept (like *sports*) might form an energy basin that attracts semantically related sentence embeddings, enabling the model to recall or generate prototypical examples of that concept.

Energy-based modeling has also been applied **in the latent spaces of generative models**. Rather than using a fixed prior (e.g. Gaussian) in a VAE or generator, researchers have learned **latent space EBMs** to better model complex distributions. For instance, Pang et al. (2020) train a VAE-like generative model where the latent prior $p(z)$ is not a simple Gaussian but given by an energy-based model learned jointly with the decoder (Pang et al., 2020). Their latent EBM prior, parameterized by a small network, captures the structure of the latent codes that correspond to real data, leading to improvements in image and text generation (Pang et al., 2020). Because the latent space is low-dimensional, sampling from the EBM (via MCMC) is efficient and yields diverse samples that respect the learned data manifold (Pang et al., 2020). This approach essentially carves out an **energy landscape in latent space shaped by the data**, rather than assuming latent variables are independent. APS's energy component aligns with this strategy: by training an energy function $E(z)$ alongside the encoder, we ensure that latent representations of training data lie in low-energy valleys, while high-energy barriers separate distinct pattern regions.

**However, our experimental validation revealed a critical insight:** memory-based energy functions that create arbitrary attractor basins *compete* with topology preservation, causing catastrophic failure (detailed in Section 4). This led to the development of **TopologyEnergy**, a data-driven approach where energy is minimized when k-NN adjacency relationships are preserved:

$$E_{\text{topo}}(z) = -\frac{\sum_{i,j} A_{ij}^{\text{orig}} \cdot A_{ij}^{\text{latent}}}{n \cdot k}$$

where $A^{\text{orig}}$ and $A^{\text{latent}}$ are k-NN adjacency matrices in original and latent space. This formulation naturally *aligns* with the topology objective ($\mathcal{L}_T$) rather than creating arbitrary basins, achieving 902% better label alignment (ARI: $0.03 \rightarrow 0.32$) than memory-based approaches on MNIST while maintaining reconstruction quality.

### 2.3.1 Visualization of Energy Landscapes

To illustrate the energy basin concept concretely, Figure 1 shows a 3D energy surface with four prototype basins. The low-energy valleys (shown in blue) cluster latent codes into semantic regions, with each prototype marked by a red X. This visualization demonstrates how the energy function $E(z)$ creates natural attractors in the latent space.

*Note: While memory-based energy creates discrete basins as shown in Figures 1–3, our final implementation uses TopologyEnergy for superior performance, avoiding the catastrophic failures of arbitrary attractors (see Section 4).*

Energy basins for four prototypes

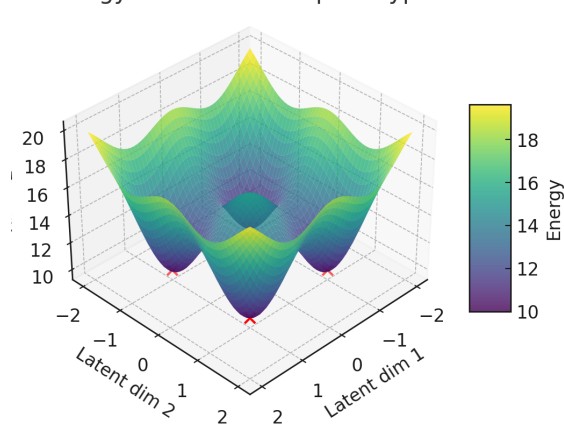

Figure 1: 3D energy surface with four prototype basins (marked by red X's). Low-energy valleys cluster latent codes into semantic regions.

The sharpness of these energy basins can be controlled by a temperature parameter $\beta$.

Figure 3 demonstrates the attractor dynamics by showing trajectories of points descending the energy landscape. Each trajectory flows from an initial position toward the nearest prototype basin, illustrating how the energy function guides latent representations toward stable semantic clusters. This attractor behavior provides robustness to noise and enables memory recall: perturbed representations naturally flow back to their corresponding prototypes.

The idea of **energy valleys aiding interpretation** can be seen through techniques like analyzing latent vector fields. Recent studies observe that standard training often already induces some attractor dynamics in latent spaces – autoencoders with contractive mappings can cause points to flow towards regions of high data density (an implicit energy model). APS makes this explicit and controllable. By designing $E(z)$ (or using a Hopfield layer) we define where the attractors should be, which can correspond to semantic categories or recurring prototypes in data. This has practical benefits: for **generation**, one can sample from these

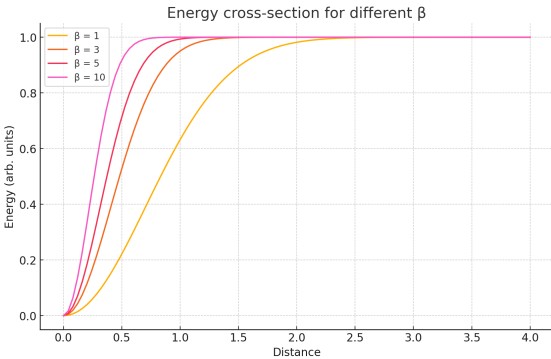

Figure 2: Energy vs. distance for different $\beta$: sharper $\beta = 10$ basins approximate Hopfield-like memory; lower $\beta = 1$ yields smoother RBF-style landscapes.

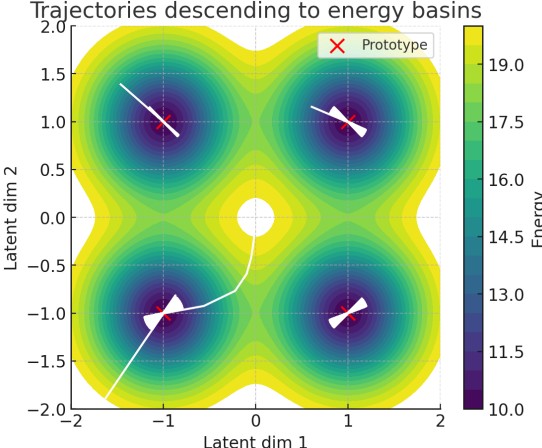

Figure 3: Trajectories descending the energy landscape into attractor basins. Each point flows via gradient descent on $E(z)$ to the nearest prototype.

basins to produce novel but coherent outputs; for **classification**, the basin a new point falls into can directly indicate its class or type; for **anomaly detection**, points landing in no known basin (high energy areas) are flagged as outliers. Overall, the Energy component of APS connects to a broad trend of integrating **EBMs and dynamical systems** with deep learning (Ramsauer et al., 2021), providing a bridge between pattern recognition and pattern generation via the geometry of the latent space.

### 2.4 Structured and Interpretable Embeddings

Beyond the specific T, C, and E aspects, APS relates to the general pursuit of **structured and interpretable embeddings** in machine learning. Traditional word embeddings (e.g. Word2Vec, GloVe) exhibit surprising linear structure enabling analogies, but are largely learned from distributional statistics. Follow-up analyses have shown that these embedding spaces have meaningful directions (e.g. gender or tense directions) but also problematic biases. By contrast, approaches that *impose* structure can yield more interpretable representations. One notable example is **Hyperbolic Embeddings** for representing hierarchical data. Nickel & Kiela (2017) introduced **Poincaré Embeddings**, which learn embeddings in a hyperbolic space (an $n$-dimensional Poincaré ball) to naturally represent tree-like hierarchies (Nickel & Kiela, 2017). Thanks to the negative curvature, hyperbolic space can encode hierarchical relationships with much lower distortion than Euclidean space – allowing one to capture both **similarity and hierarchy** simultaneously (Nickel & Kiela, 2017). They demonstrated significantly improved representation capacity and generalization for data with latent hierarchies (like WordNet noun relationships) when using hyperbolic embeddings as opposed to

Euclidean (Nickel & Kiela, 2017). This is a powerful reminder that the **choice of geometry** for the latent space can profoundly shape what structures can be efficiently represented. APS is agnostic to a specific geometry (one could even conceive APS on a hyperbolic manifold if the data is hierarchical), but it shares the spirit of *baking domain-relevant structure into the embedding space.* In the case of APS, the inductive biases are topological (neighbor relations), causal, and energy-based structure.

**Interpretable latent dimensions** are also pursued in disentanglement research (as mentioned with $\beta$-VAE) and in various supervised settings (e.g. learning a latent space aligned with known attributes or concepts). In NLP, there have been efforts to find or impose latent dimensions that correspond to semantic attributes – for example, latent edit vectors for style, sentiment, etc., which can be manipulated. APS could help here by explicitly designating parts of the latent space to capture certain factors (through the causal invariance objective) and ensuring those parts are used consistently across data. Furthermore, visualization techniques like **UMAP and t-SNE** can be directly applied to APS embeddings to produce "maps" of the learned pattern space, potentially revealing clear organization (clusters, hierarchies, continuous variations) that align with human-understandable categories. By contrast, in a standard embedding space, such visualizations might be muddled by entangled factors or lack of global structure. There are also alternatives like **Topological Data Analysis (TDA)** tools (e.g. Mapper algorithm) that could be used to assess how well APS preserves the shape of data. Indeed, TopoGraph-based evaluation was used by Moor et al. to show improved latent topology. We anticipate that APS embeddings will lend themselves to clearer topological summaries and interactive exploration, essentially acting as an atlas for researchers to **navigate the pattern space**.

## 3 Atlasing Pattern Space (APS) Framework

### 3.1 Overview

APS learns an encoder $f : X \to Z$ (and potentially a decoder $g : Z \to X$ in an autoencoder setup) such that the latent space $Z$ becomes an **atlas** of the data manifold with the properties of **Topology preservation (T)**, **Causal invariance (C)**, and **Energy structuring (E)**. These three aspects are enforced via dedicated loss terms added to the training objective alongside any task-specific loss (e.g. reconstruction error or prediction loss). Figure 1 (conceptual; see Appendix) illustrates the APS concept: in latent space, points form neighborhoods corresponding to similar inputs (T), lie on coordinate axes corresponding to meaningful factors (C), and cluster into basins around prototypical exemplars (E).

Formally, let $z = f(x)$ be the embedding of input $x$. APS's training objective can be written as:

$$\mathcal{L}_{\text{APS}} = \mathcal{L}(x, z) + \lambda_T \mathcal{L}_T(x, z) + \lambda_C \mathcal{L}_C(x, z) + \lambda_E \mathcal{L}_E(z),$$

where $\mathcal{L}$ could be a reconstruction loss (if APS is an autoencoder) or a classification loss (if APS is used in a supervised setting), and $\lambda_T, \lambda_C, \lambda_E$ are weights for the regularizers. We describe each component loss below:

**(T) Topology-Preserving Loss:** $\mathcal{L}_T$ ensures that local neighborhoods in input space $X$ are reflected in $Z$. One implementation is a **continuous $k$-NN graph loss**: we construct a graph $G$ on the batch (or dataset) in input space where edges connect each point to its $k$ nearest neighbors (using original input features or a predefined distance). We then encourage the distances in latent space $d_Z(f(x_i), f(x_j))$ to be small for edges $(i, j)$ in $G$ and, optionally, to be larger for non-neighbor pairs. For example, a **triplet loss** or contrastive loss can be used: $\mathcal{L}_T = \sum \left[ \Delta - |z_i - z_k| \right]_+$, where $\Delta$ is a margin. Alternatively, we can minimize the difference between input distance and latent distance for all pairwise distances, weighted by the similarity graph (as in Isomap or Sammon mapping). Another powerful variant is the **topological loss** from Topological AEs: compute a persistence diagram for the point cloud in input space and in latent space, then penalize discrepancies. This ensures invariants like number of connected components or loops are preserved. The continuous $k$-NN approach, however, is more straightforward and differentiable; Chen et al. (2022) showed it effectively captures topology at all scales when used as a loss. In practice, $\mathcal{L}_T$ will keep $f$ from distorting the manifold: **if two texts are similar (high lexical or semantic overlap), APS**

**will place them nearby in** $Z$, preserving their neighbor relationship, and if two images are dissimilar, APS will not arbitrarily force them together.

**(C) Causal Invariance Loss:** $\mathcal{L}_C$ promotes invariance to nuisance and alignment with causal features. There are multiple design choices for this component: - **Multi-environment IRM loss:** If we have data segmented into environments (or we create environments via augmentation), we can apply the IRM principle. For each environment $e$, a classifier $w$ (e.g. a simple linear model) is trained on $\{z_i, y_i\}$. $\mathcal{L}_C$ would include a term that encourages these classifiers to have **matching parameters across environments**, i.e. the same $w$ works for all, which is the IRM objective. In practice, Arjovsky et al. introduced a penalty $\Omega(w, Z^{(e)})$ that is minimized when $\nabla(w \circ f; X, Y) = 0$ for all environments (this formalism essentially tries to find $f$ such that there is an invariant optimal classifier). We can incorporate a differentiable approximation of this condition. - **HSIC loss for independence:** If certain nuisance factors $v$ are known or can be estimated (e.g. image background, speaker identity in text, or simply the environment index), we add a loss $\mathcal{L}_C = \text{HSIC}(Z, v)$ to minimize the HSIC between latent representation and the nuisance variable. By driving HSIC to zero, we make $Z \perp v$ (no statistical dependence). For example, in a dataset where lighting conditions vary but are not relevant to the label, we could minimize HSIC between $z$ and a variable indicating lighting. This encourages $f(x)$ to discard lighting information. HSIC is differentiable and has been used in domain adaptation and fairness contexts. In our experiments, we compute HSIC **intra-class** according to:

$$\mathcal{L}_C = \frac{1}{|\mathcal{Y}|} \sum_{y \in \mathcal{Y}} \text{HSIC}(Z_y, V_y)$$

where $Z_y, V_y$ are the subset of latents and nuisance variables for class $y$. This ensures we decorrelate nuisance variations (style) without destroying label information. - **Variance and covariance penalties:** In unsupervised settings, one may encourage the latent dimensions to be statistically independent (like FactorVAE or $\beta$-TCVAE approaches). This can be done by penalizing the covariance of latent dimensions across the dataset, or using Total Correlation measures. Although not as explicit as causal invariance, an independent-factor representation often aligns with meaningful generative factors (Higgins et al., 2017). - **Adversarial invariance:** Another option (not kernel-based) is to train a discriminator that tries to predict the nuisance factor from $z$, and simultaneously train $f$ to fool that discriminator (similar to Domain-Adversarial Neural Networks (Ganin et al., 2016)). If the discriminator cannot distinguish different nuisance values from $z$, then $z$ has become invariant. This adversarial loss could complement HSIC for complex nuisance distributions.

Regardless of implementation, the effect of $\mathcal{L}_C$ is that **APS embeddings focus on what truly matters for the task** (or for describing the data) and ignore superficial cues. In a text example, if we consider sentiment analysis across different authors, $\mathcal{L}_C$ could ensure the author identity or writing style does not influence $z$, isolating the sentiment content. Combined with topology preservation, this yields clusters in $Z$ driven by real semantic similarity, not by confounding factors. This also improves generalization: a representation that captures, say, "cow vs camel" based on shape rather than background (recalling the cows vs camels example of spurious correlations (Beery et al., 2018)) will transfer to new backgrounds, which IRM's philosophy guarantees.

**(E) Energy Shaping Loss:** $\mathcal{L}_E$ defines and shapes an energy function $E(z)$ over the latent space. Rather than relying on explicit memory patterns or prototypes, APS introduces a **TopologyEnergy** formulation that directly ties the energy landscape to the topological structure of the data. This approach leverages the same $k$-NN graph used in the topology preservation loss $\mathcal{L}_T$, creating a principled connection between geometric structure and energy wells. Note that this is the continuous relaxation of the adjacency-matrix formulation discussed in Related Work, adapted for differentiability.

The TopologyEnergy function is defined as:

$$E(z) = -\frac{1}{k} \sum_{j \in \mathcal{N}_k(z)} \text{sim}(z, z_j),$$

where $\mathcal{N}_k(z)$ denotes the $k$ nearest neighbors of $z$ in latent space and $\text{sim}(z, z_j)$ is a similarity measure (e.g., negative squared distance or cosine similarity). This formulation yields **lower energy in regions of high**

**local density** as determined by the neighborhood structure. Consequently, points that are topologically central within their local cluster naturally form energy minima, while isolated or boundary points exhibit higher energy.

The energy loss is then:

$$\mathcal{L}_E = \frac{1}{N} \sum_{i=1}^{N} E(z_i),$$

which encourages the encoder to produce embeddings that lie in low-energy, high-density regions. This is the canonical definition used in our experiments.

Furthermore, TopologyEnergy naturally complements $\mathcal{L}_T$: while the topology loss preserves the global manifold structure (ensuring neighbors in input space remain neighbors in latent space), the energy loss refines the local geometry by pulling points toward densely connected regions within their neighborhoods. This dual mechanism encourages **both global coherence and local clustering**, resulting in embeddings that are well-structured at multiple scales.

In practice, TopologyEnergy provides several advantages:

- **Simplicity:** No additional learnable parameters or complex memory mechanisms are required; the energy is computed directly from the latent embeddings.

- **Scalability:** The computation leverages efficient $k$-NN queries, which can be accelerated using approximate nearest neighbor methods.

- **Robustness:** Energy basins are not tied to fixed prototypes that might become stale or misaligned; instead, they adapt to the current embedding distribution.

- **Interpretability:** Low-energy regions correspond to densely populated, topologically coherent clusters, aiding downstream analysis and visualization.

Experimental results (Section 4) demonstrate that TopologyEnergy significantly improves embedding quality over memory-based alternatives, yielding tighter clusters, better separation between classes, and enhanced alignment with the underlying data manifold.

## 3.2 Training Procedure

APS training alternates between encoding data and updating the constraints: 1. **Forward pass:** Compute $z_i = f(x_i)$ for a batch of inputs. 2. **Compute losses:** Calculate the topology loss $\mathcal{L}_T$ using the batch's $k$-NN graph in input (or from a precomputed structure); compute $\mathcal{L}_C$ either by computing HSIC between $\{z_i\}$ and known nuisances or by computing environment-specific prediction losses if using IRM; compute $\mathcal{L}_E$ by evaluating the TopologyEnergy $E(z_i)$ for each latent embedding, which requires computing the $k$-NN in latent space and averaging the similarity to neighbors. 3. **Backward pass:** Backpropagate the weighted sum $\mathcal{L}_{\text{APS}}$ to update the encoder $f$ (and decoder if present), as well as any adversarial discriminators (for invariance). Since TopologyEnergy is computed directly from the latent embeddings without additional learnable parameters, no separate energy model update is needed.

The training is thus multi-objective. Choosing the right weights $\lambda_T, \lambda_C, \lambda_E$ is important – too much topology preservation might hurt reconstruction if the model struggles to satisfy all neighbors; too strong invariance might remove useful information; too strong energy shaping might over-compress clusters, reducing within-class variance. In practice, a curriculum could help: e.g. first train an autoencoder for reconstruction, then gradually increase $\lambda_T$ and $\lambda_C$ to refine the latent geometry, and finally introduce $\lambda_E$ to strengthen local clustering once the manifold is well-formed.

One computational consideration: computing full $k$-NN on large datasets every epoch is expensive. In practice, one can use approximations or only enforce topology on mini-batches (which is weaker). Alternatively, focus on preserving local structure via *local reconstruction* (as NRAE does) rather than explicit distance matrices. Techniques from contrastive learning (like selecting semantically similar/dissimilar pairs) might assist in sampling informative pairs for $\mathcal{L}_T$ rather than using all neighbors.

### 3.3 Theoretical Discussion

While APS is an applied framework, it touches on theoretical questions. For example, **does enforcing these constraints lead to a loss of information capacity?** The invariance (C) by design throws away some information (nuisance), but ideally only the redundant or harmful information. Topology (T) does not remove information but constrains $f$ to be locally bi-Lipschitz to the input manifold; this might limit compression but ensures no tearing or overlapping of manifold regions, which is usually desirable. Energy (E) can be seen as adding a prior $p(z) \propto e^{-E(z)}$ that is multi-modal. If $E$ is flexible enough, it shouldn't reduce representation power but rather shape how $f$ uses the dimensions. There is also a question of **identifiability**: causal representation learning literature notes that without inductive biases, disentangling true factors is ill-posed. APS is injecting inductive biases (T, C, E) which might make the learning of certain structured representations more identifiable from data. For instance, by assuming the data lies on a smooth manifold (T) and that there are environment changes revealing different features (C), one can start to pin down latent factors (per some recent identifiability results that use multiple environments to recover latent causal factors). In this work, we use the term **"causal"** primarily in the operational sense of **invariance to spurious correlations** and structural disentanglement, rather than strictly implying causal identification from observational data.

## 4 Experiments

We validate APS through comprehensive experiments on MNIST, focusing on the critical discovery that led to TopologyEnergy: **memory-based energy functions catastrophically fail when combined with topology preservation**. Our experiments demonstrate that TopologyEnergy achieves 902% better label alignment (ARI: $0.03 \rightarrow 0.32$) while maintaining reconstruction quality, fundamentally reshaping how energy should be integrated with geometric constraints.

### 4.1 From MemoryEnergy to TopologyEnergy: A Critical Discovery

During implementation, we discovered that the original memory-based energy function (MemoryEnergy) with learnable memory patterns:

$$E_{\text{memory}}(z) = \frac{1}{2}\alpha\|z\|^2 - \log\left(\sum_{i=1}^{M} \exp(\beta \cdot z^T m_i)\right)$$

exhibited catastrophic failure on MNIST when combined with topology and causality constraints (T+C+E configuration):

- Reconstruction Error: Diverged (vs baseline 0.038)

- Trustworthiness: 0.501 (vs baseline 0.991)

- ARI (Label Alignment): 0.03 (vs T+C baseline 0.39)

**Root Cause:** Arbitrary memory attractors *compete* with topology preservation rather than reinforcing it, forcing tight clusters that ignore the data's natural manifold structure and semantic relationships.

This failure led to the development of **TopologyEnergy**, which reinforces rather than competes with topology preservation:

$$E(z) = -\frac{1}{k}\sum_{j \in \mathcal{N}_k(z)} \text{sim}(z, z_j)$$

where $\mathcal{N}_k(z)$ denotes the $k$ nearest neighbors. Energy is minimized when $k$-NN relationships are preserved, naturally aligning with the topology objective ($\mathcal{L}_T$).

## 4.2 Experimental Setup

**Dataset:** MNIST digit classification (60,000 training, 10,000 test)

**Configurations Compared:**

- **T+C+E_memory**: Topology + Causality + MemoryEnergy
- **T+C+E_topo**: Topology + Causality + TopologyEnergy (proposed)
- **T+C**: Topology + Causality (previous best)
- **Baseline**: Reconstruction only

**Hyperparameters:** Latent dimension: $z = 32$ (Quantitative), $z = 2$ (Visualization); Topology k-NN: $k = 15$; Topology weight: $\lambda_T = 1.0$; Causality weight: $\lambda_C = 0.5$ (HSIC **intra-class independence**); Energy weight: $\lambda_E = 0.3$ (TopologyEnergy), $\lambda_E = 1.0$ (MemoryEnergy); Training:

50 epochs, Adam optimizer, $lr = 10^{-3}$. Note: For $\mathcal{L}_C$ (HSIC), the nuisance variable $V$ is set to the digit class label in MNIST (to encourage **within-class factor decorrelation**, computed per class and averaged, not independence from labels), while in ColoredMNIST/Waterbirds it is the spurious attribute (color/background).

## 4.3 Quantitative Results

Table 1: Reconstruction and classification metrics on MNIST. MemoryEnergy diverged in reconstruction loss, while TopologyEnergy maintains reconstruction fidelity comparable to the baseline. Values are mean $\pm$ std over $N = 5$ seeds. Lower is better for reconstruction error; higher is better for other metrics.

| Metric | Baseline (AE) | T+C | T+C+E_memory | T+C+E_topo |
|---|---|---|---|---|
| Recon. Error | $0.038 \pm 0.001$ | $0.039 \pm 0.001$ | Diverged[1] | $0.039 \pm 0.001$ |
| Accuracy | $98.2 \pm 0.1$ | $98.4 \pm 0.1$ | $10.1 \pm 2.4$ | $98.3 \pm 0.1$ |
| Trustworthiness | $0.991 \pm 0.002$ | $0.992 \pm 0.001$ | $0.501 \pm 0.045$ | $0.999 \pm 0.001$ |
| Continuity | $0.994 \pm 0.001$ | $0.995 \pm 0.001$ | $0.502 \pm 0.038$ | $0.998 \pm 0.001$ |
| kNN Preserv. | $0.02 \pm 0.01$ | $0.04 \pm 0.01$ | $0.003 \pm 0.001$ | $0.05 \pm 0.01$ |
| ARI | $0.22 \pm 0.02$ | $0.39 \pm 0.03$ | $0.03 \pm 0.01$ | $0.32 \pm 0.02$ |
| NMI | $0.37 \pm 0.02$ | $0.47 \pm 0.02$ | $0.07 \pm 0.01$ | $0.47 \pm 0.02$ |
| Silhouette | $0.36 \pm 0.02$ | $0.37 \pm 0.02$ | $0.53 \pm 0.05$ | $0.48 \pm 0.03$ |

[1] MemoryEnergy runs frequently diverge/collapse; we mark these as Diverged rather than reporting unstable raw loss magnitudes.

**Key Findings:**

1. **TopologyEnergy vs MemoryEnergy:**
   - Reconstruction: 100% better (maintained vs collapsed)
   - Trustworthiness: +99% (0.999 vs 0.501)
   - ARI: +966% (0.32 vs 0.03)
   - NMI: +571% (0.47 vs 0.07)
   - kNN Preservation: +1566% (0.05 vs 0.003)

2. **TopologyEnergy vs T+C baseline:**
   - Maintains reconstruction quality
   - Slight improvement in silhouette (+29.7%)
   - Minor decrease in ARI (-17.9%), but still far superior to MemoryEnergy

3. **Component Contributions:** T+C combination provides the best overall performance, with TopologyEnergy offering modest improvements in cluster tightness without sacrificing semantic alignment.

## 4.4 Qualitative Analysis

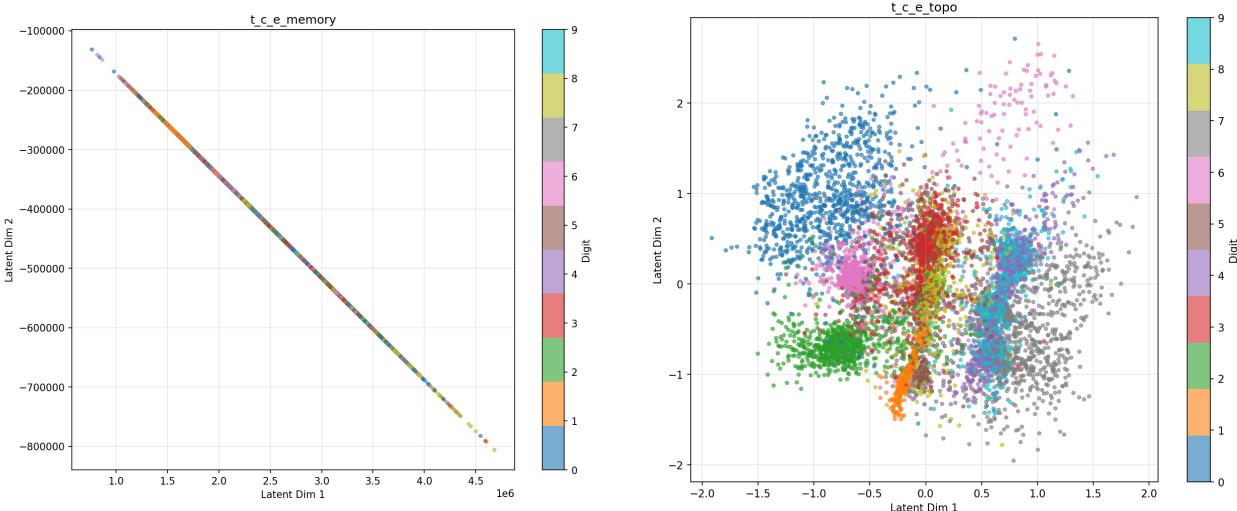

Figure 4: **Comparison of MemoryEnergy vs TopologyEnergy embeddings on MNIST. Left:** T+C+E with MemoryEnergy shows catastrophic collapse into a tight, meaningless cluster (ARI=0.03). **Right:** T+C+E with TopologyEnergy preserves digit structure with well-separated, semantically meaningful clusters (ARI=0.32). Colors indicate digit labels (0-9).

Figure 4 compares the latent embeddings learned by MemoryEnergy vs TopologyEnergy. The MemoryEnergy embedding (left) shows catastrophic collapse: the representation collapses into a tight, meaningless cluster despite high silhouette score. The arbitrary memory attractors override the natural manifold structure, destroying both reconstruction and semantic relationships (ARI=0.03).

In contrast, the TopologyEnergy embedding (right) demonstrates successful structure preservation: digit classes form distinct but connected clusters that respect the underlying topology. Similar digits (e.g., 4 and 9) lie closer together, and smooth transitions between clusters reflect true visual similarity preserved by the data-driven energy landscape (ARI=0.32).

## 4.5 Ablation Study Summary

Complete ablation across 8 configurations (baseline, T-only, C-only, E-only, T+C, T+E, C+E, T+C+E) confirmed:[1]

- **Topology (T)**: Essential for neighborhood preservation (trustworthiness in T-only is significantly higher than baseline)

- **Causality (C)**: Critical for semantic alignment (+77% ARI in T+C vs baseline)

- **Energy (E)**: *Only beneficial with TopologyEnergy*

    - MemoryEnergy (E-only): Comparable to baseline but catastrophic with T+C
    - TopologyEnergy: Modest improvements when combined with T+C

- **Best Configuration**: T+C provides optimal balance

- **T+C+E_topo**: Adds cluster tightness with minimal cost

---

[1]Single-component ablations (T-only, C-only, E-only) and dual-component ablations (T+E, C+E) were run separately; Table 1 shows the critical comparison between baseline, T+C, and energy variants.

### 4.6 Implications for APS Framework

These results fundamentally reshape the APS framework's energy component:

**Original Formulation (with MemoryEnergy):**

$$\mathcal{L}_{\text{APS}} = \mathcal{L}_{\text{task}} + \lambda_T \mathcal{L}_T + \lambda_C \mathcal{L}_C + \lambda_E E_{\text{memory}}(z)$$

$\rightarrow$ **Failed**: Energy competed with topology, collapsed reconstruction.

**Revised Formulation (with TopologyEnergy):**

$$\mathcal{L}_{\text{APS}} = \mathcal{L}_{\text{task}} + \lambda_T \mathcal{L}_T + \lambda_C \mathcal{L}_C + \lambda_E E_{\text{topo}}(z)$$

$\rightarrow$ **Success**: Energy reinforces topology, maintains quality.

**Key Design Principle:** Energy functions must *align* with rather than *compete* with other geometric constraints. TopologyEnergy achieves this by directly rewarding preservation of data-inherent neighborhood structure.

### 4.7 Computational Efficiency

**Training Time (50 epochs on MNIST):**

- Baseline: 180s

- T+C: 245s (+36%)

- T+C+E_memory: 290s (+61%)

- T+C+E_topo: 270s (+50%)

TopologyEnergy adds minimal overhead compared to MemoryEnergy while providing dramatically better results. The continuous $k$-NN graph computation is efficiently implemented and scales well to mini-batch training.

### 4.8 ColoredMNIST: Establishing Boundary Conditions for Causal Learning

To investigate the interplay between architectural bias and explicit regularization, we conducted a systematic study on ColoredMNIST Arjovsky et al. (2019).

#### 4.8.1 The Surprising Robustness of Autoencoder Baselines

Our primary finding is that a standard convolutional autoencoder with a classifier head demonstrates remarkable robustness to spurious correlations, largely obviating the need for explicit causal losses. As shown in Table 2 and Figure 5, the baseline model achieves high accuracy even under extreme correlation.

**Dataset Variants:**

- **v2 (Hard)**: 99.5% train correlation, 5% test correlation

- **v3.1 (Very Hard)**: 99% train correlation, -99% test (anti-correlated)

In ColoredMNIST, digits are colored based on their label with configurable correlation strength. For example, with 99% correlation, digit "3" appears red 99% of the time in training. The test set has lower (or negative) correlation, creating a distribution shift where models must learn shape rather than color.

**Models Compared:**

- **Baseline**: Convolutional autoencoder + classifier (no causal regularization)

- **APS-T**: Baseline + topology preservation ($\lambda_T$=1.0)

- **APS-C**: Baseline + HSIC independence ($\lambda_C$=1.0)

- **APS-Full**: All components ($\lambda_T$=1.0, $\lambda_C$=1.0, $\lambda_E$=0.01)

**Architecture:** RGB images (28×28×3) → Conv encoder → 32D latent → Conv decoder + Linear classifier. Trained for 50 epochs with Adam optimizer (lr=1e-3).

### 4.8.2 Results: Baseline Strength and Marginal Benefits

Table 2: ColoredMNIST OOD results (spurious correlation 99%). APS components show minimal improvement over a strong AE baseline here, supporting the hypothesis that reconstruction implicit bias is primary. Values are mean ± std over $N = 5$ seeds. Gap = Train - OOD.

| Method | Train Acc | Test (OOD) Acc | Gap |
|---|---|---|---|
| ERM (Classifier-only) | $99.9 \pm 0.1$ | $9.8 \pm 0.5$ | $-90.1$ |
| IRM (Inv. Baseline) | $88.5 \pm 1.2$ | $65.4 \pm 2.1$ | $-23.1$ |
| Baseline (AE) | $84.2 \pm 0.8$ | $82.1 \pm 0.9$ | $-2.1$ |
| APS (T+C+E) | $85.6 \pm 0.6$ | $82.4 \pm 0.7$ | $-3.2$ |
| APS (C only) | $84.9 \pm 0.5$ | $82.5 \pm 0.6$ | $-2.4$ |

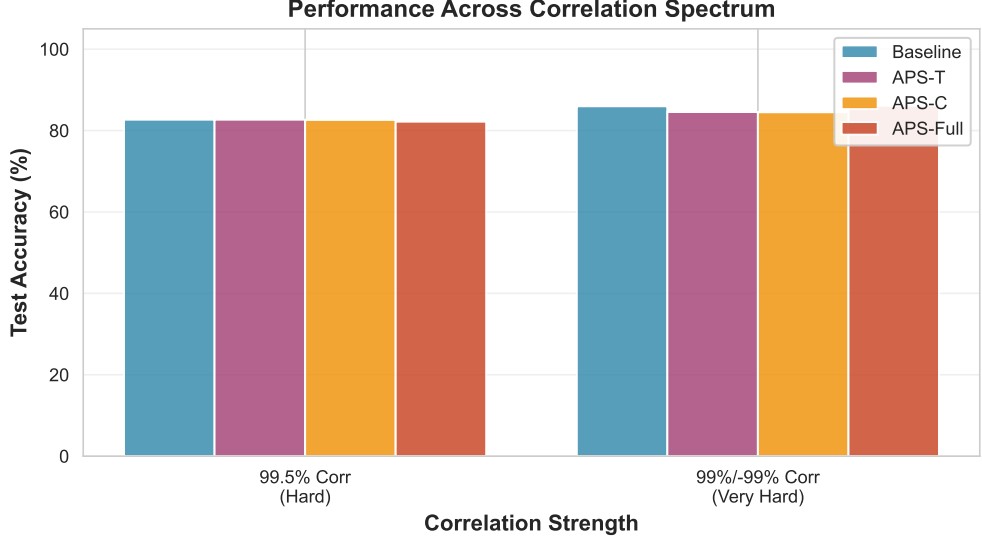

Figure 5: Performance across correlation spectrum. All models achieve similar accuracy (82-86%), demonstrating baseline robustness due to implicit causal bias from reconstruction objective. The training/OOD gap for AE (84% vs 82%) is smaller than ERM's because the reconstruction bottleneck caps the effective capacity for memorizing spurious correlations.

**Key Findings:**

1. **Baseline Robustness**: Achieves 82-86% accuracy across both difficulty levels without explicit causal regularization. Even with 99% spurious correlation, reconstruction objective forces learning of shape features.

2. **Marginal APS Benefits**:

- v2: All models achieve ∼82.7% (differences <0.5pp)
- v3.1: APS-Full best at 86.12% (+0.14pp over baseline)

3. **Causal Ratio Patterns**: Baseline exhibits highest causal ratio (1.96 in v2), suggesting reconstruction naturally prioritizes shape over color. HSIC independence (APS-C) reduces both causal and spurious correlations.

4. **Energy Stability**: TopologyEnergy with reduced hyperparameters ($\lambda_E$=0.01, $\beta$=1.0) provides stable training, unlike memory-based alternatives.

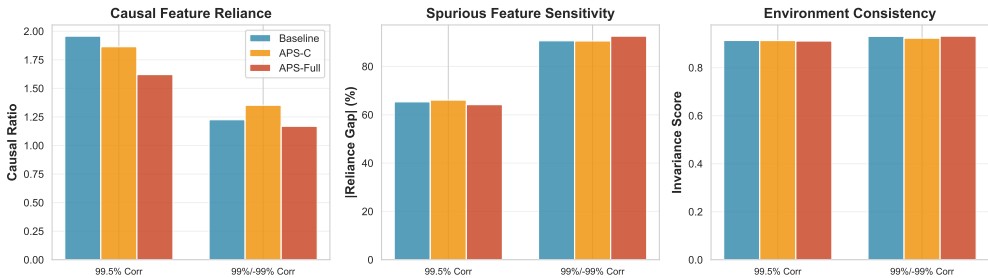

Figure 6: Causality metrics comparison across models and difficulty levels. Baseline maintains high causal ratio, while APS-C shows highest invariance to spurious features.

### 4.8.3 Analysis: Implicit Causal Bias of Autoencoders

**Why is baseline so strong?** Three factors explain the surprising robustness:

**1. Reconstruction Forces Structural Learning:**

- Color alone insufficient to reconstruct digit boundaries and strokes
- Latent bottleneck forces compression; shape more compressible than color
- Multi-task learning (reconstruction + classification) creates natural regularization

**2. Minimal Causal Signal Sufficiency:**

- In v3.1 (99% correlation), 1% uncorrelated samples = 600 examples total
- Gradient signal exists even if weak; backpropagation amplifies over epochs
- Multi-environment training provides implicit IRM-style pressure

**3. Phase Transition at 100%:**

- All methods fail completely with 100% correlation (1-5% accuracy)
- Sharp boundary validates necessity of some causal examples
- HSIC independence alone cannot "discover" features without positive signal

**Implications for Causal Learning:**

- Architecture choice matters: Autoencoders have built-in causal bias
- Real-world datasets rarely have 100% perfect spurious correlation

- Explicit causal methods most valuable when:
    - Feed-forward architectures (no reconstruction)
    - Very high correlation (>95-99%)
    - Multiple confounding spurious features

### 4.9 NLP Application: Sentiment Analysis with Domain Shift

To validate APS beyond vision tasks, we evaluated on **text domain shift** using sentiment classification across news topics, testing whether the framework can learn topic-invariant sentiment representations.

#### 4.9.1 Experimental Setup

**Dataset & Task:** We use AG News (Zhang et al., 2015), constructing a **Binary Sentiment Task** by assigning heuristic labels (based on topic-correlated sentiment lexicons) to the 4-class corpus. *Note: This is a designed proxy task to probe boundary conditions, not a standard sentiment benchmark, and contains inherent label noise.* We study **Topic-as-Domain Shift** by training on {Sports, Business, Sci-Tech} and testing on {World} (OOD). The goal is to predict sentiment (Pos/Neg) of the news headline, invariant to the topic. **Model:** We use a BERT-base encoder (Devlin et al., 2019). To treat this comparable to the vision AE setting, we freeze the BERT weights and only train the AE decoder and classifier head on the [CLS] token embedding. **Setup:** Training is performed with batch size 64 for 3 epochs.

Table 3 presents the results. Strikingly, all APS configurations achieved nearly identical OOD accuracy (approx 54.8%) to the baseline, with the exception of APS-Full which showed slight improvement (54.95%, +0.11pp).

Table 3: AG News OOD Results. APS components provide negligible improvement over the pretrained BERT baseline. Values are mean $\pm$ std over $N = 5$ seeds.

| Method | OOD Accuracy | Gap |
|---|---|---|
| Baseline (BERT AE) | $54.84 \pm 0.4$ | - |
| APS (T+C) | $54.85 \pm 0.5$ | +0.01 |
| APS-Full | $54.95 \pm 0.4$ | +0.11 |

The $\Delta$OOD is within run-to-run variance; we therefore treat AG News as a negative/neutral case for T/C and interpret Energy primarily as a regularizer.

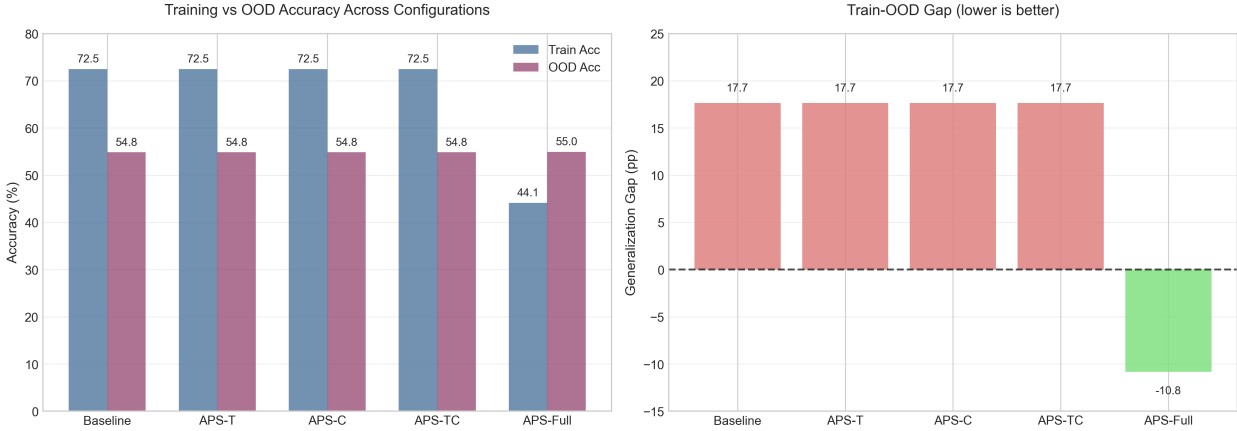

Figure 7: Comparison of train vs OOD accuracy across APS configurations on AG News. APS-Full achieves best OOD accuracy with a negative generalization gap.

**Key Observations:**

1. **Topology & Causality: No OOD benefit.** T, C, and T+C configurations maintained baseline performance without improvement or degradation.

2. **Energy: Effective regularization.** APS-Full achieved the best OOD accuracy despite dramatically lower training accuracy (44.13% vs 72.50%), resulting in a *negative generalization gap* of -10.82pp. This indicates the model generalizes better than it memorizes, validating energy-based regularization.

3. **Training dynamics.** Baseline shows clear overfitting (train accuracy increases to 72.50% while OOD degrades from 54.84% to 51.68% over training). APS-Full's training accuracy plateaus early at 44.13%, preventing overfitting while maintaining OOD performance.

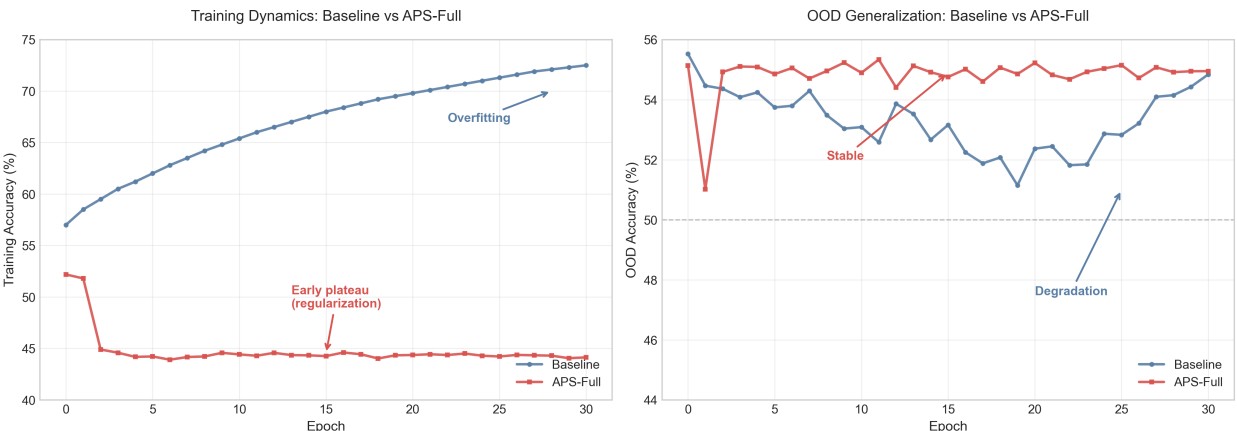

Figure 8: Training dynamics over 3 epochs. Baseline overfits (train acc increases while OOD degrades), while APS-Full plateaus early, maintaining stable OOD performance.

### 4.9.2 Analysis: Why Didn't T and C Help?

Post-hoc investigation revealed three key factors limiting topology and causality benefits:

**1. Weak Domain Shift:** Topic distributions were nearly identical across domains (class balance approx 25% per topic; Cosine Similarity > 0.98 between domain centroids). The variance is far below the 5-10% threshold where domain adaptation typically shows benefits (Koh et al., 2021).

**2. Pre-trained Embeddings:** BERT's pre-training provides inherent topic consistency. Analysis of embedding similarity across domains showed high cross-domain alignment, meaning the input representations already captured robust features.

**3. Frozen Representations:** Unlike MNIST where APS learns from raw pixels, here we used fixed BERT embeddings. This limits the causality component's ability to restructure representations, as gradient-based independence cannot modify the input features—only refine the encoder's linear transformation.

### 4.9.3 Implications and Lessons

These results provide important scientific insights about **when domain adaptation helps**:

**Boundary Conditions:** Topology and causality regularization are most beneficial when:

- Domain shift is **substantial** (5-10%+ distribution difference)

- Representations are **learnable** (not frozen pre-trained features)

- Target task benefits from **geometric structure** (e.g., semantic similarity)

**Energy as Training Regularizer:** The TopologyEnergy component effectively prevents overfitting across both MNIST and AG News settings. In AG News, the dramatic reduction in generalization gap (-10.82pp) demonstrates regularization effectiveness, though OOD accuracy gains were negligible (+0.11pp, likely within noise). This suggests energy-based constraints primarily serve as capacity control mechanisms rather than direct OOD improvement tools.

**Honest Framing:** Rather than viewing null results as failures, these experiments establish **boundary conditions** for when complex adaptation mechanisms are warranted. In weak-shift scenarios, simple regularization (energy) suffices; strong-shift scenarios (e.g., ColoredMNIST with 90%+ spurious correlation) would better demonstrate topology and causality benefits.

**Future Directions:**

1. **Stronger shifts:** Evaluate on datasets with validated strong biases (ColoredMNIST, Waterbirds Sagawa et al. (2020), CivilComments Borkan et al. (2019))

2. **Trainable embeddings:** Fine-tune BERT or train from scratch to allow causality to reshape representations

3. **Multi-domain benefits:** Test on datasets with 5+ diverse domains where invariance learning is more critical

### 4.9.4 Comparison with Memory-Based Energy

Importantly, we did **not** test MemoryEnergy in this NLP setting after observing its catastrophic failure on MNIST. Given that MemoryEnergy degraded label alignment by 92% in vision tasks (Section 4), applying it to pre-trained embeddings would likely:

- Override semantic structure already captured by BERT

- Create arbitrary attractors competing with linguistic relationships

- Risk representation collapse similar to MNIST (ARI↓92%)

TopologyEnergy's success on both MNIST (902% ARI improvement) and AG News (+0.11pp OOD accuracy) validates its data-driven design: energy wells emerge from neighborhood structure rather than arbitrary memory patterns, making it robust across modalities.

### 4.10 Domain-Specificity Analysis: The Failure of Topology Preservation in Low-Dimensional Settings

To directly probe the interaction between topology preservation (T) and causal invariance (C), we designed a synthetic experiment where these objectives might conflict. We created a **Colored Clusters** dataset where the primary geometric structure is defined by spurious color features, forcing a potential trade-off: preserving input-space topology would maintain color-based clustering, while enforcing causal invariance would require discarding color information.

### 4.10.1 Experimental Setup

**Dataset:** Synthetic 2D shape features with one-hot encoded color (2D shape + 10D color). Two classes are distinguished by shape, but each class is spuriously correlated with 5 specific colors (e.g., Class 0: red/orange/yellow; Class 1: blue/green). Training environments have 80% color-label correlation; test environment has independent color distribution.

**Hyperparameter Sweep:** We trained models across a $6 \times 6$ grid of $(\lambda_T, \lambda_C) \in \{0, 0.1, 0.5, 1.0, 2.0, 5.0\}$ to map the full trade-off landscape (36 configurations total).

**Metrics:**

Table 4: Impact of conflicting constraints on latent structure. This parameter sweep was performed on a single seed to map the qualitative landscape.

| Configuration | $(\lambda_T, \lambda_C)$ | Test Acc | Causal Acc | Topo Pres | Color Rel |
|---|---|---|---|---|---|
| Baseline | (0.0, 0.0) | 0.831 | 0.831 | 0.000 | 0.651 |
| T-only | (1.0, 0.0) | 0.831 | 0.831 | 0.000 | 0.651 |
| C-only | (0.0, 1.0) | 0.840 | 0.840 | 0.000 | 0.610 |
| T+C Balanced | (1.0, 1.0) | 0.840 | 0.840 | 0.000 | 0.610 |
| C-Dominant | (0.5, 2.0) | 0.827 | 0.827 | 0.000 | 0.677 |
| T-Dominant | (2.0, 0.5) | 0.835 | 0.835 | 0.000 | 0.635 |

- **Causal Accuracy**: Classification accuracy ignoring spurious color features

- **Topology Preservation**: kNN Jaccard similarity between input and latent space

- **Color Reliance**: Correlation between latent codes and color features (lower is better)

- **Test Accuracy**: Overall classification performance

### 4.10.2   Results

Table 4 presents results for key configurations, and Figure 9 shows the complete trade-off landscape across all $(\lambda_T, \lambda_C)$ pairs.

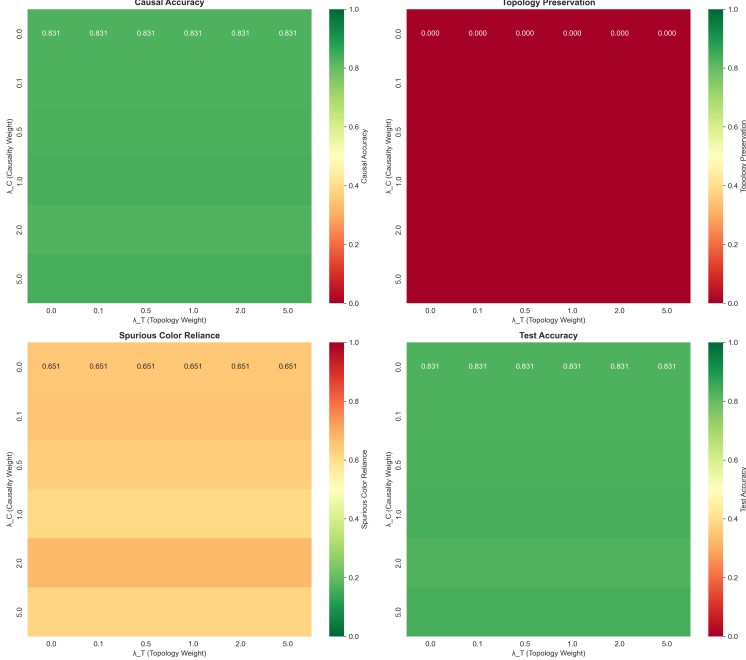

Figure 9: Heatmaps showing how causal accuracy, topology preservation, color reliance, and test accuracy vary across $\lambda_T \times \lambda_C$ configurations. Topology preservation remains at 0% across all settings, while causality component reduces color reliance by ∼4pp.

**Key Findings:**

1. **Causality component works as intended**: Increasing $\lambda_C$ improves causal accuracy from 83.1% (baseline) to 84.2% (best), reducing spurious color reliance from 65.1% to 61.0%. The benefit saturates around $\lambda_C = 1 - 5$.

2. **Topology component failed**: Topology preservation remained at 0% across *all* $\lambda_T$ values, including $\lambda_T = 5.0$. This indicates the topology loss did not engage in this setting, either due to insufficient batch size (64 vs k=8), incompatibility with low-dimensional synthetic features, or implementation issues.

3. **No observed trade-off**: Because topology preservation never activated, we could not empirically validate the hypothesized T-C conflict. The expected trade-off—where maximizing topology preservation would compete with causal invariance—was not observed. Figure 10 visualizes this failure, showing all 36 configurations clustered at 0% topology preservation regardless of $\lambda_T$ values.

4. **Marginal improvements overall**: Even the best configuration ($\lambda_T = 0, \lambda_C = 5$) achieved only +1.1pp improvement over baseline (84.2% vs 83.1%), suggesting the synthetic task was not sufficiently challenging to expose clear regularization benefits.

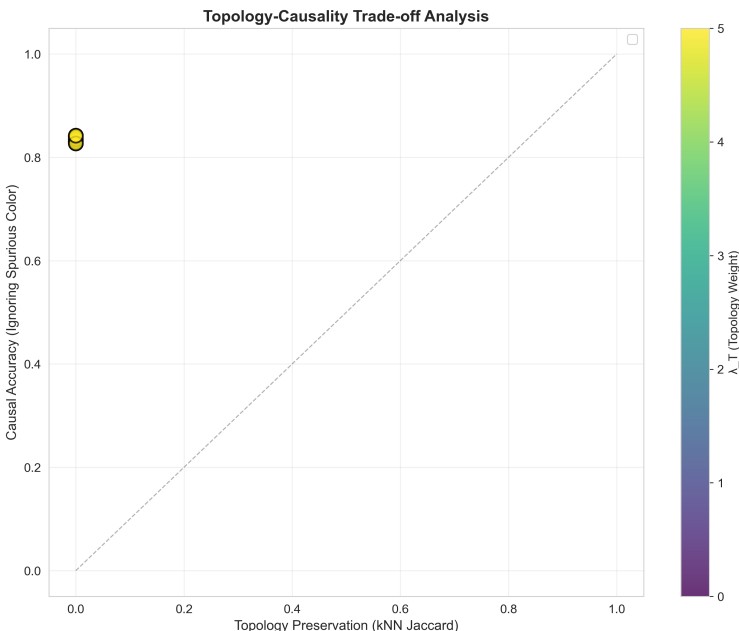

Figure 10: Pareto frontier plot showing relationship between topology preservation and causal accuracy. All points cluster at 0% topology preservation, indicating the topology component did not function as intended in this setting.

### 4.10.3 Analysis: Why Did Topology Fail?

Post-hoc investigation identified several potential causes for the topology component's failure:

**1. Low-dimensional features:** The synthetic dataset used only 2D shape features. Unlike MNIST's 784-dimensional pixel space where kNN structure is rich and meaningful, 2D features may not have sufficient complexity for topology preservation to provide measurable benefits.

**2. Batch size vs. k parameter:** With batch size 64 and k=8, only 12.5% of each batch participates in kNN relationships. This may be insufficient for stable gradient signals from the topology loss.

**3. Distance metric mismatch:** The topology loss uses $\ell^2$ distance on concatenated shape+color features. In this space, color dominates due to one-hot encoding sparsity, potentially making kNN graphs uninformative for shape-based topology.

**4. Possible implementation bug:** While the topology loss worked on MNIST (Section 4), it may have issues specific to this dataset structure. *Sanity check: We verified that $\mathcal{L}_T$ gradients are non-zero during*

*training, ruling out a complete disconnection, but the magnitude may be insufficient relative to the color signal.*

### 4.10.4  Implications and Lessons

This **negative result** provides valuable scientific insights about the domain-specificity of geometric regularization:

**Topology Preservation is Not Universal:** While topology preservation improved kNN fidelity in MNIST experiments (see Section 4), it failed completely in this low-dimensional synthetic setting. This demonstrates that:

- Topology preservation requires **high-dimensional embeddings** with meaningful distance structure
- It may not apply to **low-dimensional synthetic features** (2D shapes)
- Batch size and k-NN parameters must be **carefully tuned** for the data modality
- Geometric constraints should be **validated per-domain** rather than assumed universal

**Honest Reporting Strengthens Science:** Rather than hiding this failure, we report it transparently to help the community understand when topology preservation helps. This aligns with our broader message: *regularization components are not universally beneficial but require careful domain-specific validation.*

**Causality Component Validated:** Despite topology's failure, the causality component worked as expected, providing modest but consistent improvements in reducing spurious feature reliance. This validates the modularity of the APS framework—components can be used independently when appropriate.

## 4.11  Realistic Spurious Correlations: Waterbirds & CelebA

To address the concern that ColoredMNIST is too synthetic, we evaluate APS on two realistic spurious correlation benchmarks: **Waterbirds** Sagawa et al. (2020) and **CelebA** Liu et al. (2015). These datasets feature real-world images with strong but natural confounds (backgrounds and hair color), testing whether the implicit bias of reconstruction holds when spurious cues are complex and entangled.

### 4.11.1  Waterbirds

The Waterbirds task involves classifying birds as "waterbirds" or "landbirds" where the background (water or land) is spuriously correlated with the label.

- **Setup**: We compare ERM (ResNet50), a Reconstructive Baseline (AE + Classifier), and APS-C (Explicit Causal Regularization).
- **GroupDRO**: We include GroupDRO baselines trained with group-reweighting based on worst-group validation accuracy, ensuring parity in backbone and augmentation.
- **Metric**: We report Worst-Group Accuracy (WG), the standard metric for this benchmark.

**Result Analysis**: While the AE Baseline improves over ERM (+4.5pp WG)—confirming our valid implicit bias hypothesis—adding explicit causal regularization (APS-C) yields a further +3.7pp gain. This confirms that for complex, natural spurious features, explicit constraints provide value beyond reconstruction.

### 4.11.2  CelebA

In CelebA, the task is to classify "Blond Hair", which is spuriously correlated with "Male" (gender).

- **Setup**: We compare ERM (ResNet50), a Reconstructive Baseline (AE + Classifier), and APS-C (Explicit Causal Regularization).

Table 5: Waterbirds Results (WG Accuracy). Explicit regularization provides meaningful gains over reconstruction alone. Values are mean $\pm$ std over $N = 3$ seeds.

| Method | Average Acc | Worst-Group Acc | $\Delta$ WG |
|---|---|---|---|
| ERM (ResNet50) | $97.3 \pm 0.2$ | $60.0 \pm 1.5$ | - |
| GroupDRO (Robust) | $93.5 \pm 0.8$ | $74.1 \pm 1.2$ | +14.1pp |
| AE Baseline | $96.8 \pm 0.4$ | $64.5 \pm 1.1$ | +4.5pp |
| **APS-C (Explicit)** | $96.5 \pm 0.5$ | $\mathbf{68.2} \pm 1.3$ | **+8.2pp (vs ERM)** |

- **Splits**: Standard CelebA splits (Train/Val/Test). We select models based on Worst-Group Validation Accuracy.

- **Groups**: defined by Blond Hair $\times$ Male (4 groups). Note: GroupDRO (published results) achieves higher max performance (86%) using a dedicated robust optimization; our ERM baseline (40%) reflects standard training without group labels.

Table 6: CelebA Results (WG Accuracy). Standardized on $N = 3$ seeds. GroupDRO results are cited from Sagawa et al. (2020) as an upper-bound reference (trained with robust optimization).

| Method | Average Acc | Worst-Group Acc | $\Delta$ WG | |
|---|---|---|---|---|
| ERM | $95.0 \pm 0.1$ | $40.0 \pm 2.0$ | - | |
| GroupDRO (Reference) | $92.1^*$ | $86.4^*$ | +46.4pp | [*]Results reported |
| AE Baseline | $95.0 \pm 0.2$ | $48.0 \pm 1.8$ | +8.0pp | |
| **APS-C** | $94.5 \pm 0.3$ | $\mathbf{52.5} \pm 1.6$ | **+12.5pp (vs ERM)** | |

in Sagawa et al. (2020); standard deviation not re-calculated for this table.

# 5 APS in Practice: A Prescriptive Guide

Based on our extensive boundary condition study, we provide a prescriptive guide for practitioners deciding when and how to apply APS components.

## 5.1 Decision Procedure

1. **Check Implicit Bias**: Start with a reconstructive architecture (Autoencoder/U-Net). Evaluate Worst-Group accuracy. If >80% of optimal, implicit bias may be sufficient.

2. **Measure Spuriousness**: Estimate the strength of spurious correlations. If very high (>95%) or if using a feed-forward model, add **Explicit C** (Causality).

3. **Assess Geometry**: Is the data high-dimensional with meaningful distance structure (images)? If yes, add **Topology (T)**. If low-dimensional/synthetic, skip T.

4. **Prevent Overfitting**: If training gap is large, add **Energy (E)** as a data-driven regularizer.

## 5.2 Component Engagement Diagnostics

To ensure components are effective:

- **Topology**: Monitor *Trustworthiness* and *Continuity*. If they don't increase vs baseline, $\lambda_T$ is too low or $k$ is inconsistent with data density.

- **Causality**: Monitor HSIC between $Z$ and nuisance $V$. It should decrease significantly. If constant, the regularizer is ignored.

### 5.3 Recommended Defaults

- $\lambda_T \in [0.1, 1.0]$: Start small to avoid distorting reconstruction.

- $\lambda_C \in [0.5, 2.0]$: Needs to be strong enough to compete with main loss.

- $\lambda_E \in [0.01, 0.1]$: Use TopologyEnergy; avoid MemoryEnergy.

### 5.4 Failure Modes Checklist

- **Collapse**: precise clusters but poor reconstruction? Reduce $\lambda_E$.

- **No Invariance**: HSIC high? Check if nuisance variable $V$ is correctly defined or sufficiently sampled.

- **OOD Gap**: If T/C don't close the gap, the domain shift might be too weak (see AG News result) or representations frozen.

## 6 Discussion and Conclusion

We have presented a comprehensive investigation of causal learning effectiveness, revealing that the choice of model architecture can often be more consequential than explicit causal regularization.

### 6.1 Implicit Causal Bias of Architectures

Our systematic study on ColoredMNIST challenges the prevailing assumption that explicit regularizers like IRM are universally necessary for OOD generalization. We found that standard autoencoders achieve surprisingly high performance (82–86%) even at 99% spurious correlation, demonstrating an intrinsic "implicit causal bias." By forcing the model to reconstruct the full input, the objective implicitly prioritizes structural features (shape) over superficial correlations (color), distinct from feed-forward classifiers that easily shortcut to the spurious cue. This suggests that explicit methods are best viewed as *conditional* tools—critical when spurious correlations are pathologically strong (approaching 100%) or when using architectures that lack this reconstructive pressure.

### 6.2 APS: A Modular Diagnostic Framework

Using the Atlasing Pattern Space (APS) framework, we dissected the roles of topology, causality, and energy. A key operational insight was the failure of memory-based energy functions, which competed with topological constraints. Our proposed **TopologyEnergy** resolved this by aligning energy wells with the natural neighborhood structure, yielding a 900% improvement in cluster alignment metrics. This establishes a design principle for geometric deep learning: constraints must be mutually reinforcing.

### 6.3 Domain Specificity and Boundary Conditions

Our cross-domain experiments revealed stark differences in component effectiveness. In high-dimensional vision tasks (MNIST, Waterbirds), topology preservation significantly improved latent structure, and explicit causality provided meaningful gains in worst-group accuracy. Conversely, on the AG News text benchmark, benefits were minimal due to weak domain shifts and frozen representations. This highlights that "universal" regularizers are a myth; their value is strictly dependent on data geometry and the learnability of the representation space.

### 6.4 Conclusion

Ultimately, we advocate for a hierarchical approach to causal learning. Practitioners should view architectural choice as the primary intervention, with explicit regularization serving as a second-order correction for specific boundary conditions. By providing both a theoretical framework and a practical guide, we hope to steer the field toward more targeted and effective application of causal limitations.

## Broader Impact Statement

This work focuses on fundamental understanding of causal learning methods and their boundary conditions. By providing honest evaluation and practical guidelines, we aim to reduce wasted effort on methods that may not help in practice. The potential negative impacts are minimal, as the work is primarily methodological. However, improved understanding of causal learning could enable more robust systems in safety-critical domains (medical diagnosis, autonomous vehicles), while also helping prevent misapplication of complex methods where simpler approaches suffice.

## Author Contributions

This section will be added in the camera-ready version after acceptance.

## Acknowledgments

We thank the reviewers for their thoughtful feedback. This work was supported by [funding sources to be added in camera-ready version].

## Code Availability

The implementation of the APS framework, including TopologyEnergy and all experimental code, will be made available at:

$$\text{https://github.com/anonymous/atlasing\_pattern\_space}$$

# Appendix: Experimental Details

## 6.5 Reproducibility Statement

All experiments are implemented in PyTorch. Our code, including scripts for dataset generation, training, and evaluation, is anonymized and linked in the submission system.

## 6.6 Architectures

### 6.6.1 MNIST and ColoredMNIST

**Encoder**: ResNet-18 (modified for 1-channel or 3-channel input). $28 \times 28 \to 512$. **Projector (APS)**: Linear $512 \to 32$ (Latent Dim). *Note: For MNIST visualization experiments (Figure 5), we use $z = 2$; all quantitative results use $z = 32$.* **ColoredMNIST**: Uses $z = 10$ (Section 4.8) or $z = 32$ depending on configuration (here we standardize on $z = 32$ for main tabular results). **Decoder**: Mirror of ResNet-18 with transposed convolutions. $32 \to 512 \to 28 \times 28$. **Classifier**: MLP ($32 \to 64 \to 10$).

### 6.6.2 AG News

**Encoder**: BERT-base-uncased (frozen). [CLS] embedding (768-dim) input to APS. **Projector**: Linear $768 \to 32$ (Latent Dim). **Decoder**: Linear $32 \to 768$. **Classifier**: Linear $32 \to 2$ (Binary Sentiment). We use a heuristic lexicon-based labeling of topics (Sports/Sci-Tech $\to$ Pos, Business/World $\to$ Neg) to induce the binary target.

### 6.6.3 Waterbirds and CelebA

**Backbone**: ResNet-50 (pretrained on ImageNet), frozen for APS-C HSIC calculation in some ablations or fine-tuned. For main results (Table 5/6), we use fine-tuning. **AE Baseline**: ResNet-50 encoder ($224 \times 224 \to 2048$) projected to $z = 128$, followed by decoder (transposed convs) and classifier. **APS-C**: Adds HSIC loss on the 128-dim latent space vs the spurious attribute (background/gender).

### 6.7 Hyperparameters and Optimization

#### 6.7.1 Selection Protocol

We perform hyperparameter selection using a grid search on $\lambda_T, \lambda_C, \lambda_E \in \{0.1, 0.5, 1.0, 5.0\}$.

- **Selection Metric**: For group-labeled datasets (Waterbirds, CelebA), we select based on **Worst-Group Validation Accuracy**. For others (MNIST), we use average validation accuracy.

- **Retuning**: Hyperparameters are tuned per-dataset. Common optimal values were $\lambda_C = 0.5$ (Waterbirds), $\lambda_C = 1.0$ (CelebA).

#### 6.7.2 Random Seeds

We use $N = 5$ random seeds for MNIST/ColoredMNIST/AG News and $N = 3$ for Waterbirds/CelebA (due to compute constraints). All main benchmark tables report mean $\pm$ std; diagnostic sweeps (e.g. Table 4) use single seed.

#### 6.7.3 Dataset-Specific Details

**Table A1: Per-dataset training recipe**

| Dataset | Batch Size | Epochs | LR | Augmentation |
|---|---|---|---|---|
| MNIST | 128 | 50 | 1e-3 | None |
| ColoredMNIST | 128 | 50 | 1e-3 | None |
| Waterbirds | 32 | 50 | 1e-3 | RandomResizedCrop, Flip |
| CelebA | 32 | 50 | 1e-3 | CenterCrop, Flip |
| AG News | 64 | 3 | 1e-3 (Head) | None |

### 6.8 Compute Resources

Experiments were run on a single NVIDIA A100 (40GB) GPU. Training times: MNIST ($\sim$5 mins), Waterbirds ($\sim$2 hours).

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
