# OpenReview forum: "When Does Causal Regularization Help? A Systematic Study of Boundary Conditions in Spurious Correlation Learning"
_TMLR — Rejected by TMLR_

### Review · Reviewer_DLau · 2025-12-15

**Summary Of Contributions:**

The authors study how to stop AI models from using misleading shortcuts (spurious correlation) and help them learn real features.
When the test data differs from the training data (out-of-distribution), models that rely on shortcuts often fail because the shortcut patterns do not hold in the test set.

They find autoencoders which learn by reconstructing the input can naturally make models to learn more real features. On ColoredMNIST, autoencoder models perform well already and adding extra causal penalties like IRM or HSIC only improves results a little.

**Strengths:**
1. Authors conduct analysis and experiment prove to show architecture choice is the key,

-Reconstructive architectures like autoencoders provide a powerful Implicit Causal bias, which naturely force the model to learn structure, not shortcuts. On ColoredMNIST, autoencoders reach 82–86% accuracy even with 99% spurious correlation.

-Explicit methods like IRM or HSIC add only small gains (0-4pp).

2. The paper makes a valuable contribution by showing that regularization components are not universally beneficial but rather require careful domain-specific validation.

-The authors argue that we should not assume a complex regularizer will always help. We should first verify that the data satisfies the regularizer’s assumptions (e.g., the data has sufficient dimensionality and meaningful geometry of the latent space for the regularizer to be effective).

-To support this argument, the authors designed specific tests to evaluate how regularization effectiveness changes across domains. In Section 4.10, they introduced a synthetic dataset with 2D shape features. They demonstrated that their Topology tool failed in this setting because the data was low-dimensional and lacked sufficient geometric structure. In contrast, they showed the same tool works well on MNIST dataset, where 784-dimensional feature gives rich dimensionality and meaningful geometry for their Topology tool to succeed.

3. The paper introduces the Atlasing Pattern Space (APS) framework. The authors emphasize that APS is designed as a diagnostic tool, not just a new algorithm. This is an important design choice:

-APS is a modular toolkit that combines topology preservation, causal invariance, and energy shaping. Each component can be turned on or off, which makes the analysis simple and clear.

-The paper does a good job to show not only when components help, but also when they interfere or fail (for example, the conflict between memory-based energy functions and topology preservation).

-The APS framework turns causal regularization into a testable hypothesis that helps explain results.

4. The paper provides empirical evidence across multiple domains, such as image (MNIST, ColoredMNIST), NLP (AG News), and synthetic datasets.

**Weakness:**

1. This paper shows in most cases, the proposed tools fail or have no measurable effect. A boundary condition study should demonstrate both where a method works well and where it breaks down. Therefore, aside from the original MNIST image task, it's better to provide a convincing win case where the proposed APS gives clear benefits.

The paper aims to provide guidance on when to use the proposed tools, but the experiments mainly demonstrate that they do not help in non-image cases.

For example:

Example 1: On the AG News NLP dataset, APS provides almost no improvement. As stated in Section 4.9.2, “all APS configurations achieved nearly identical OOD accuracy (54.84%) to the baseline, with the exception of APS-Full which showed slight improvement (54.95%, +0.11pp).”

Example 2: On the synthetic dataset with 2D shape features, the APS Topology component fails. As mentioned in Section 4.10.2, “topology preservation remained at 0% across all 36 different settings.”

It would strengthen the paper to include at least one convincing win case in which the proposed APS framework provides clear and meaningful benefits.

2. The paper argues "explicit causal regularization is unnecessary". However, I have a concern that this conclusion could only be true for simple dataset like ColoredMNIST, where reconstruction naturally encourages the model to focus on digit shape.

In real-world tasks (e.g., medical imaging or social media analysis), spurious cues might be mixed with the true signal. In those cases, reconstructing the input might not be enough to separate real features from spurious correlation.

It would strengthen the claim if also tested on a more realistic dataset with complex spurious correlations, and showed that the same conclusion still holds.

3. The proposed APS framework appears complex (e.g., multiple loss terms, k-NN graph, HSIC, energy functions, and careful hyperparameter tuning), but it provides negligible gains over simpler baselines.

Similar to Weakness 1., the experiments largely show that APS gives little or no benefit in non-image cases. It would strengthen the paper to include at least one convincing win case in which APS delivers clear, meaningful improvements (not just tiny changes) over standard baselines.

**Audience:**

Yes

**Audience Explanation:**

This paper would be of interest to TMLR's audience working on causal representation learning, spurious correlations, and out-of-distribution generalization.

**Claims And Evidence:**

Yes

**Claims Explanation:**

The experiments are quite carefully designed and clearly support the claim that architectural choice is crucial, and that regularization components are not universally beneficial but require careful, domain-specific validation.

It would further strengthen the paper to include at least one convincing win case where APS yields clear, meaningful improvements (rather than only tiny changes) over standard baselines, especially beyond vision tasks. Most current results only show that APS provides little or no benefit in non-image settings.

**Requested Changes:**

Please kindly refer to Weakness 1., 2., and 3.

---

> ### Author Response · Authors · 2026-01-02
> **Response to Reviewer DLau**
>
> Thank you for the positive assessment of the core idea and the clear requests for stronger win cases and more realistic validation.
> (1) “Need at least one convincing win case where APS gives clear benefits.”
> * Resolved: We added realistic benchmarks where explicit constraints show meaningful gains on Worst-Group accuracy:
>     * Waterbirds: APS-C improves WG beyond AE baseline; includes trained GroupDRO comparator (Section 4.11.1; Table 5).
>     * CelebA: APS-C improves WG beyond AE baseline; GroupDRO provided as a robust literature reference (Section 4.11.2; Table 6).
> (2) “Conclusion may only hold for simple datasets like ColoredMNIST… test on realistic datasets with complex spurious correlations.”
> * Resolved: Added Waterbirds and CelebA specifically to test entangled confounds, and we position ColoredMNIST as a controlled boundary-condition probe rather than sole evidence (Section 4.11; Tables 5–6; Discussion).
> (3) “APS seems complex but provides negligible gains, especially beyond vision.”
> * Partially resolved:
>     * We now show a clear regime where APS-C improves WG in realistic vision benchmarks (Section 4.11).
>     * For NLP, we keep AG News as a boundary-condition probe and interpret near-null T/C gains as evidence of weak engagement under frozen representations and weak shift (Section 4.9; Table 3).
>     * We added Section 5 to make APS usage more efficient and reduce unnecessary complexity via decision rules and diagnostics.
> * Future work: expand beyond AG News to standard text bias benchmarks (e.g., CivilComments/WILDS) to test APS in stronger NLP spurious-correlation regimes.

---

### Review · Reviewer_Qwxs · 2025-12-15

**Summary Of Contributions:**

In this seemingly unfinished paper draft, the authors present the APS framework for training classifiers that are resistant to spurious correlation information. They train an auto encoder on the input data and potentially add further regularisation, which rarely seems to help much though. They evaluate their method on coloured MNIST a sentiment analysis on AG NEWS and a synthetic low-d dataset. Generally, the results are not particularly impressive, based on very small datasets and appear to be mostly caused by the auto encoder, when they happen. Also, the authors do not provide many important details about these tests, like the architecture of the classification network for example. Also there are no baselines or comparisons anywhere in this draft which makes the given evaluations mostly useless.

My main criticism of this paper remains that it is just not finished: There are two citation styles mixed in here and only a small part of the citations are listed in the reference list. The discussion section consists mostly of bullet points. The performance numbers are inconsistent across the paper. Quite a few details of the experiments are missing, etc.

**Additional Comments:**

The authors provide a GitHub link in the paper that appears to be not defaced breaking blind review to some extend. I did not investigate to remain reasonably blind myself, but a GitHub account name is definitely not anonymous.

**Audience:**

Yes

**Audience Explanation:**

If this was true, some people might be interested in this. This is very hard to judge though, as there is so much information missing.

**Broader Impact Concerns:**

No that seems fine.

**Claims And Evidence:**

No

**Claims Explanation:**

Given the unfinished state of this manuscript, it is hard to judge whether there is anything there. Even if I take everything at face value, I do not think the provided experiments support the statements made in the abstract and introduction.

**Requested Changes:**

Absolutely critical would be finishing the writing of this paper. References, experiment descriptions, & the discussion are all clearly unfinished and this is just not a paper at all yet.

Then the authors should provide some baseline comparisons to let us know how good their method actually is.

I am not sure that the results developed here will be enough in the end to actually support a scientific insight.

---

> ### Author Response · Authors · 2026-01-02
> **Response to Reviewer Qwxs**
>
> Thank you for the genuine feedback. the revised vesrion directly addresses the “unfinished manuscript” concerns and adds the missing baselines and experimental details.
> (1) “This is a seemingly unfinished paper draft… references, experiment descriptions, & discussion are unfinished.”
> * Resolved: revised version includes a full editorial pass:
>     * citation style unified; complete references,
>     * discussion rewritten in full prose (no bullet-only sections),
>     * inconsistent numbers and formatting issues removed.
> (2) “No baselines or comparisons… evaluations mostly useless.”
> * Resolved: Added standard baselines and consolidated them into the main results:
>     * ColoredMNIST: ERM (classifier-only) + IRM + AE baseline + APS variants (Section 4.8; Table 2).
>     * Waterbirds: includes trained GroupDRO (Section 4.11.1; Table 5).
>     * CelebA: includes GroupDRO as a clearly marked literature reference (Section 4.11.2; Table 6).
> (3) “Missing important details… architecture… etc.”
> * Resolved: Added a structured reproducibility appendix:
>     * Reproducibility statement + code availability (Appendix 6.5),
>     * architectures (Appendix 6.6),
>     * hyperparameters + selection protocol (Appendix 6.7.1),
>     * seed policy (Appendix 6.7.2),
>     * per-dataset recipe summary (Table A1).
> (4) “GitHub link appears not anonymous (breaks blind review).”
> * Resolved: We provide an anonymized code repository and explicitly state anonymization in the paper (Code Availability / Appendix 6.5).

---

### Review · Reviewer_FGRQ · 2025-12-26

**Summary Of Contributions:**

The paper challenges a widely held thought in causal and OOD learning that explicit causal regularization is necessary to prevent models from relying on spurious correlations. This work shows that reconstructive architectures like autoencoders has a strong implicit causal bias, enabling robust OOD generalization on ColoredMNIST even under extreme spurious correlation. In this case, explicit causal regularizers provide only marginal gains, which suggest that architectural inductive bias can dominate explicit causal constraints. In order to formalize it, the authors introduce Atlasing Pattern Space, a modular framework that decomposes inductive bias into three orthogonal components. Another major contribution is the identification of boundary conditions under which each component is effective. Via experiments on MNIST, ColoredMNIST, AG News, and synthetic datasets, the paper shows that explicit causal regularization becomes critical only when architectural bias is weak or causal signal is nearly absent. Also, topology preservation benefits high-dimensional continuous domains but fails completely in low-dimensional synthetic settings.

Pros:
1. This paper proposes an interestig and new direction in casual learning rather than proposing yet another regularizer.
2. The ColoredMNIST results are compelling and carefully analyzed.
3. This work emphasizes when methods fail and honestly reports negative results.
4. APS serves as a useful lens for disentangling different inductive biases, even if not fully adopted as a new standard method.

Cons:
1. While ColoredMNIST is a standard testbed, it is highly synthetic. The paper’s strongest claims rest heavily on this dataset, and evidence on more realistic spurious-correlation benchmarks is absent.
2. APS is more diagnostic than prescriptive. More instructions how to use APS in practice would be beneficial.

**Audience:**

Yes

**Audience Explanation:**

This work focuses on an interesitng topic about conditions of different methods in OOD generalization and causal regularization, combining theoretical systems and numerical verification (across vision, NLP, and synthetic settings). The analysis is systematic and reveals the situations when certain casual regularization techniques works. Someone focusing on casual inference/casual ML might be inspired from this work.

**Broader Impact Concerns:**

No ethical issues.

**Claims And Evidence:**

Yes

**Claims Explanation:**

The authors provide convincing evidence on ColoredMNIST that reconstructive architectures exhibit a strong implicit causal bias, which is showed through carefully controlled correlation regimes, ablation studies, and complementary causal diagnostics. It also includes a well characterized phase transition at 100% spurious correlation. Claims about the marginal utility of explicit causal regularization in the presence of strong architectural bias are also reasonably supported by consistent, albeit small, performance differences across multiple configurations. But as mentioned in the contribution summary, the evidence base is limited to a small set of benchmarks and causal methods, which constrains generality.

**Requested Changes:**

The primary requested changes focus on responding the previously mentioned weaknesses.

---

> ### Author Response · Authors · 2026-01-02
> **Response to Reviewer FGRQ**
>
> Thank you for the thoughtful review and constructive suggestions. We address your two main concerns below.
> (1) “ColoredMNIST is highly synthetic… evidence on more realistic spurious-correlation benchmarks is absent.”
> * Resolved: We added realistic spurious-correlation benchmarks with Worst-Group accuracy:
>     * Waterbirds: ERM vs AE baseline vs APS-C vs trained GroupDRO (Section 4.11.1; Table 5).
>     * CelebA: ERM vs AE baseline vs APS-C; GroupDRO included as a clearly marked literature reference (Section 4.11.2; Table 6).
> * These additions provide an explicit “win regime” beyond ColoredMNIST where explicit constraints improve WG performance.
> (2) “APS is more diagnostic than prescriptive… more instructions how to use APS in practice would be beneficial.”
> * Resolved: Added Section 5 “APS in Practice” including:
>     * a decision procedure (architecture-first; diagnose spuriousness; choose T/C/E),
>     * component engagement diagnostics (how to tell if T/C/E is actually active),
>     * recommended defaults, and
>     * a failure-mode checklist (when APS components should be skipped).

---

### Author Response · Authors · 2026-01-02
**Summary of Revisions**

We substantially revised the manuscript to address reviewer feedback on external validity, baselines, reproducibility, and actionability.
* Realistic benchmarks + WG evaluation: Added Waterbirds and CelebA experiments and report Worst-Group (WG) accuracy alongside average accuracy (Section 4.11; Tables 5–6). Included GroupDRO as a standard robust baseline (trained on Waterbirds; CelebA GroupDRO clearly marked as a literature reference).
* Stronger baselines/controls on ColoredMNIST: Consolidated main results to include ERM (classifier-only) and IRM alongside the reconstructive baseline and APS variants (Section 4.8; Table 2), directly testing the boundary condition “architecture absent → explicit invariance helps.”
* Mechanism-driven APS refinement (Energy): Documented catastrophic failure of MemoryEnergy under T+C+E and introduced TopologyEnergy as a principled fix (Section 4.1–4.7; Table 1; Fig. 4). Clarified the canonical TopologyEnergy used in experiments and its relation to the adjacency-matrix view.
* Actionability (“APS in Practice”): Added a prescriptive guide with a decision procedure, component engagement diagnostics, recommended defaults, and failure-mode checklist (Section 5).
* Reproducibility/reporting: Added a reproducibility statement, per-dataset architectures/hyperparameters, selection protocol, and seed policy (Appendix 6.5–6.7; Table A1). Main benchmark tables report mean ± std over seeds; Table 4 is explicitly a single-seed diagnostic sweep.
* Polish + claim tightening: Fixed citation/format issues, removed inconsistencies, and reframed claims to emphasize boundary conditions (not universal necessity). Clarified that “causal” is used in the operational invariance-to-spurious sense rather than causal identification.
* Anonymity: Provided an anonymized code repository in accordance with double-blind requirements (Code Availability / Appendix 6.5).

---

### Decision · Action_Editor_TkDJ · 2026-02-10

**Recommendation:** Reject

**Audience:**

Yes

**Audience Explanation:**

Yes. the topic studies in this paper is interesting and important to the ML research community.

**Claims And Evidence:**

No

**Claims Explanation:**

I thank the authors for their effort in improving the writing and adding the CelebA and Waterbirds experiments, as the paper has clearly improved following the reviewers' comments. However, the reviewers remain split (reject, leaning accept, and accept), and I share the negative recommendation's concerns regarding the manuscript’s current state. Specifically, the fact that pages 11–22 still consist of bullet points and figures rather than formal scientific prose, combined with the near-identical numbers reported in Figure 7 and etc, makes it difficult to establish the technical reliability of the results in its current form.

On a technical level, the +0.14pp gain for APS-Full on ColoredMNIST is likely statistical noise and sensitive to hyperparameter tuning rather than a robust algorithmic improvement. Most importantly, the core claim that causal penalties offer only tiny gains (likely within statistical noise) relies on the toy ColoredMNIST benchmark and is actually contradicted by the significant gains seen in the new real-world data (CelebA and Waterbirds), this seems to be a bit concerning. Finally, there is no statistically significant evidence that APS works in NLP (AG News).

**Resubmission Of Major Revision:**

The authors may consider submitting a major revision at a later time.